# OpenReview forum: "Active Fine-Tuning of Multi-Task Policies"
_ICML.cc/2025/Conference — ICML 2025 poster_

### Official Review · Reviewer_GTTR · 2025-03-09

**Overall Recommendation:** 3

**Summary:**

The paper tries to tackle the problem of maximizing the multi-task performance of a pre-trained policy with minimal additional demonstrations via active learning. The proposed algorithm builds upon existing active learning approaches in non-sequential domain. AMF selects queries that maximizes the expected info gain about expert policy over the occupancy with some guarantee that the resulting policy will converge to the expert policy under some assumptions. AMF proposes practical approaches to compute conditional entropy with Gaussian Process policies and occupancy estimation via importance sampling. Lastly, AMF proposes a simple technique using a prior policy to mitigate issues of catastrophic forgetting of learned behavior. The authors demonstrate the approach on a 2D didactic problem along with continuous control tasks in MetaWorld and Franka Kitchen. Further, they show that AMF can scale to work with pretrained policies such as diffusion policies.

**Claims And Evidence:**

The claims are supported with experiments in a didactic environment and continuous control experiments across several robot learning benchmarks. It would be nice to see other baseline methods that use Bayesian active selection in a decision-making context. Currently, the authors only compare against the naive baseline of uniform sampling for the next task id. The results make sense intuitively in that if there is a mismatch in distribution between the pretraining and deployment time task distributions then uniform sampling would be worse as you can query for task demonstrations suboptimally. The experimental results do support this as we observe that in the case of mismatched distributions, AMF learns quicker compared to the uniform sampling baseline.

**Essential References Not Discussed:**

There are some works in active learning for sequential decision-making and Bayesian experiment design that should have been referenced and used as potential baseline methods.
The second work also similarly proposes a Bayesian approach for query selection in the setting of model-based RL and has a similar objective for estimating occupancy measure between trajectory and expert policy. Also, they propose similar idea of using Gaussian process for modeling the policy. It would be nice to see some additional discussions contrasting to this prior work.

[1] Neiswanger, Willie, et al. "Generalizing bayesian optimization with decision-theoretic entropies." Advances in Neural Information Processing Systems 2022.
[2] Mehta, Viraj, et al. "An experimental design perspective on model-based reinforcement learning." International Conference on Representation Learning 2022.

**Experimental Designs Or Analyses:**

The experimental design seems reasonable. The authors evaluated on both state- and image- based settings. A policy is pretrained on set number of demonstrations and some tasks are heldout to simulate a distribution mismatch.

**Methods And Evaluation Criteria:**

Yes, the proposed methods make sense. The primary evaluation criteria is success rate of the learned policy over consecutive rounds of active querying of expert demonstrations. The benchmarks used in the experiments are also standard manipulation tasks used in the literature.

**Other Comments Or Suggestions:**

- I would recommend putting a topline score for what an expert policy would have achieved if provided all the training demonstrations from the very beginning without applying any active learning.
- It is not clear from the main results how much the adaptive prior is helping to prevent catastrophic forgetting.

**Other Strengths And Weaknesses:**

Strengths:
- Paper is well written and easy to follow
- Method is technically sound
- Experimental setting is reasonable and AMF is evaluated on a good suite of environments
- Results on pretrained off-the-shelf policies seem to suggest that AMF is useful for improving policy performance quickly in a more realistic setting

Weaknesses:
- Additional references expected, see above section. There are some works in active query selection for RL that are not referenced.
- It would be nice to have one additional baseline method that is a stronger baseline that uniform sampling
- Some of the ideas in the AMF algorithm have already been proposed in prior work, e.g. occupancy estimation in [2] and prior regularization for catastrophic forgetting

**Questions For Authors:**

- Why is Figure 2 showing the results at 40 demonstrations and not 50 demonstrations which is the number of demonstrations reported in the text?
- For my understanding: in the state-based experiments, if one new demonstration is added at each iteration and if there are 20 iterations, does that mean 20 new demonstrations are added? I'm quite surprised that AMF is able to learn a multi-task policy for all those tasks with so few demonstrations.
- Would this be a reasonable baseline: first evaluate the base policy to get a initial estimate of the policy's performance on each task and use that as your prior distribution to query for demonstrations? This baseline would aim to select tasks that the base policy performs poorly on.
- Are the result plots averaged across each of the tasks in the benchmark? How do I interpret the standard deviation intervals? It would be interesting to see the individual task successes with more demonstrations.

**Relation To Broader Scientific Literature:**

This paper is relevant to the multi-task learning, active imitation learning, and imitation learning literature. AMF extends the ideas of active query selection in Bayesian optimization to the sequential decision-making domain and specifically for the setting of multi-task learning.

**Theoretical Claims:**

I did not check the correctness of the proofs.

---

> ### Author Rebuttal · Authors · 2025-03-31
>
> We would like to thank the reviewer for the thorough evaluation of our submission, and the insightful comments.
>
> **Additional baselines and prior work**
>
> We thank the reviewer for suggesting these two additional works. We found them to be very relevant to this direction, but not directly applicable as baselines. To the best of our knowledge, Bayesian active selection of demonstrations for fine-tuning remains relatively unexplored. [1] focuses on a standard, non-sequential BO setting, and [2] is designed for exploration in MBRL. Nevertheless, the objective from [2] could be applied to our setting, and would select tasks inducing trajectories minimizing posterior uncertainty over dynamics predictions along optimal trajectories. This criterion would however require access to a dynamics model estimated on pre-training data. In realistic settings, pre-training data is not available; if it was, other strategies (such as rebalancing, see Appendix I) would become promising. We focus on the realistic setting in which pre-training data is not available: in this case, a dynamics model trained from scratch would be unaware of the pre-trained policy’s uncertainty, and would fail to drive effective task selection. These two references remain very relevant; we are happy to include and extend this discussion in our related works.
>
> **Adding a topline score**
>
> While we are not allowed to submit a revision at the moment, we have run the experiments to estimate a topline score. Under no mismatch, a policy fine-tuned “at once” on 20 demonstrations uniformly distributed across tasks matches the score of the Uniform baseline at iteration 20: \~64% in Metaworld and \~85% in Kitchen. If the number of demonstrations is increased 5x to 100, the policy reaches a performance of \~92% in Metaworld, and \~89% in Kitchen.
>
> **Role of the adaptive prior**
>
> We found the adaptive prior to be very helpful when pre-training and evaluation task distributions do not match. In this situation, when the agent requests the first demonstrations and fine-tunes on them, it may quickly forget tasks for which no demonstration was queried. As a consequence, the average success rate drops significantly in early iterations under mismatch (Figure 4, first and third columns, red line). This phenomenon is substantially alleviated if an adaptive prior is used (yellow line), as it can retain information about pre-training tasks. We hope this explains the issue, and would ask the reviewer whether anything else could be clarified.
>
> **Number of demonstrations in Figure 2**
>
> Thank you for catching this! We only collect 40 demonstrations, as performance plateaus afterwards. We will correct this in the text.
>
> **Data efficiency with 20 demonstrations**
>
> We directed significant efforts at making the implementation of the underlying algorithm as data efficient as possible, in order to reduce runtime. As a result, the policy performs well with only 20 additional fine-tuning demonstrations, as the reviewer points out. We must however remark that policies are already pre-trained on a number of demonstrations (\~10-20), and that, while good performance is achieved quickly, a larger number of demonstrations is required to reach asymptotic performance.
>
> **Additional baseline evaluating policy performance**
>
> Evaluating the multi-task policy, and sampling demonstration according to the inverse of the policy performance on each task would indeed represent an interesting solution to active multi-task finetuning. This is however an online solution, which has additional requirements. First, it requires access to the environment in order to estimate the policy performance, potentially involving the execution of unsafe actions. Second, it requires designing a per-task reward to evaluate the rollouts. In settings in which these issues would not be relevant (e.g. a good simulator and reward function are available), this method would do very well. In contrast, AMF is a fully offline method, which avoids environment interaction and uses uncertainty as a proxy for performance. This relationship is formally described in Theorem A.10.
>
> **Average task success curves and standard deviations**
>
> As the reviewer points out, plots are averaged across tasks. Shaded areas are 90% simple bootstrap confidence intervals of the average task performance (thus modeling stochasticity across seeds).
> Asymptotic single-task success rates on Metaworld and Kitchen converge between 70% and 100%, depending on the task. We are happy to report this in detail given a chance to update the paper.
>
> ---
> We would like to thank the reviewer for taking the time to evaluate our submission. We hope we were able to address all comments, and we would like to further discuss any point that remains unresolved.
>
> **References**
>
> 1) Neiswanger et al., Generalizing bayesian optimization with decision-theoretic entropies, NeurIPS 2022
>
> 2) Mehta et al., An experimental design perspective on model-based reinforcement learning, ICLR 2022

---

### Official Review · Reviewer_uqQh · 2025-03-10

**Overall Recommendation:** 3

**Summary:**

This paper investigates an active multi-task fine-tuning scheme, which adaptively selects the task to be demonstrated for sample-efficient fine-tuning of multi-task behavioral cloning policies. The authors provide a practical version of the proposed method and highly the efficacy of the proposed method through experiments on a controlled 2D integrator environment as well as more realistic scenarios such as Metaworld, Franka Kitchen, and the WidowX (Appendix F) environments.

**Claims And Evidence:**

Yes

**Essential References Not Discussed:**

I do not follow the relevant work in this area very closely.

**Experimental Designs Or Analyses:**

The experimental setting makes sense for evaluating the sample-efficient fine-tuning of pretrained policies through the proposed method.

**Methods And Evaluation Criteria:**

Yes

**Other Comments Or Suggestions:**

- Based on the experimental environments, does this method tackle the scenario where the environment remains the same and the policy on fine-tuned on new tasks that are introduced during evaluation? Or will this work even when the pretrained policy is introduced in a new environment with similar tasks? I am curious the hear about the authors’ thoughts on this.

**Other Strengths And Weaknesses:**

Strengths
- The authors provide theoretical guarantees of the method and provide a practical algorithm applicable to more realistic settings.
- The paper includes experiments on a controlled 2D integrator environment as well as more realistic scenarios such as Metaworld, Franka Kitchen, and the WidowX (Appendix F) environments.
- The authors study uncertain estimation choices for AMF and study its applicability to other off-the-shelf models such as Diffusion Policy (Sec 5.4) and Octo (Appendix F).
- The paper includes the limitations of the proposed method.

Weaknesses
- It would be great if the authors could include comparisons with other methods in the domain in addition to the uniform sampling baseline. This would help the reader get a sense of where the proposed method stands in comparison with existing methods.
- In Sec. 5.2, when the authors allow 10 or 20 evaluation iterations, is this across the 4 or 5 tasks in the task set? If yes, this seems like a small number of demonstrations. Does the performance in Figure 4 improve with more iterations?
- Also, since the finetuned policies get around or greater than 50% success rate despite starting with a low success rate (Fig. 4), especially in the mismatched case, does this mean that the tasks are very simple? Since 10 or 20 demonstrations across 4 or 5 tasks lead to a significant improvement in performance, even with uniform sampling.

**Questions For Authors:**

It would be great if the authors could address the weaknesses as well as the question mentioned earlier.

**Relation To Broader Scientific Literature:**

I do not follow the relevant work in this area very closely. It would be nice if the authors could compare with existing methods in the area to show where the proposed method stands compared to them.

**Theoretical Claims:**

Not in detail. The empirical results look promising.

---

> ### Author Rebuttal · Authors · 2025-03-31
>
> We would like to thank the reviewer for thoroughly reviewing our work. We are happy to address each comment below.
>
> **Comparison with other methods in the domain**
> > It would be great if the authors could include comparisons with other methods in the domain
>
> To the best of our knowledge, the problem of active-finetuning for behavior cloning has not been explored in the past. For this reason, there are no established baselines for this specific problem. For instance, [1] studies a dataset may be “curated” according to mutual information, but it does not address online data selection or multi-task learning. [2] deploys a related objective to ours in the context of model-based RL; however, this objective cannot be evaluated in our setting, as it would require a dynamics model estimated on pre-training data, which are not available during fine-tuning. The uniform baseline is what is practically used in an offline setting [3].
>
> **Number of iterations in Figure 4**
> > In Sec. 5.2, when the authors allow 10 or 20 evaluation iterations, is this across the 4 or 5 tasks in the task set?[...] Does the performance in Figure 4 improve with more iterations?
>
> In Sec. 5.2, 20 demonstrations are collected across all tasks for all plots: in non-visual settings, we collect 1 demonstration for 20 iterations, and in visual settings, we collect 2 demonstrations for 10 iterations. This is a relatively small number, but we must consider that the policy is already pre-trained on several demonstrations (~10-20 depending on the setting). Furthermore, significant implementation efforts were directed towards data-efficiency in the underlying (BC) algorithm to reduce runtime. Albeit more slowly, performance further improves with more iterations, and eventually converges to a value of ~92% in Metaworld, and ~89% in Kitchen.
>
> While Kitchen and Metaworld remain high-dimensional, challenging deep RL benchmarks, we find that they require relatively few demonstrations. Thus, we also provide an evaluation on Robomimic (Figure 6), which involves arguably harder tasks, requiring a larger number of demonstration is necessary to achieve comparable success rates (>100 demonstrations for some tasks). We find that results in this setting confirm the previous ones on Kitchen and Metaworld.
>
> **New environment or new tasks**
> > [...] will this work even when the pretrained policy is introduced in a new environment with similar tasks?
>
> Thanks for the interesting question. Our formal analysis of the algorithm assumes a fixed MDP, and thus tackles the scenario in which the environment does not change between pre-training and evaluation. However, the practical algorithm may also be applied in case of slight shifts in the environment, as long as the policy generalizes across these changes. If the environment at evaluation is completely unrelated to pre-training data, then AMF would in practice recover a less informed strategy. However, if the changes in the environment are relatively small, and uncertainty estimates remain somewhat meaningful, AMF would still be able to leverage them to efficiently query demonstrations.
>
> ---
> We again thank the reviewer for their insightful feedback. If there are any additional comments, we would be very happy to engage in further discussion.
>
> **References**
>
> [1] Hejna et al., Robot Data Curation with Mutual Information Estimators, arXiv:2502.08623
>
> [2] Mehta et al., An experimental design perspective on model-based reinforcement learning, ICLR 2022
>
> [3] Kim et al., Fine-Tuning Vision-Language-Action Models: Optimizing Speed and Success, arXiv:2502.19645

---

### Official Review · Reviewer_jtjj · 2025-03-12

**Overall Recommendation:** 3

**Summary:**

Pretrained generalist policies are becoming popular in the robot learning field for the gained capabilities by large-scale training. Nevertheless, deploying such policies in a zero-shot manner is still lacking. Hence, adaptation of generalist policies is a must to utilize the acquired representations and skills. A popular approach for adaptation is fine-tuning on some expert demonstrations. Since collecting such demonstrations is expensive, we should minimize the number of the demonstrations needed for the downstream fine-tuning. The problem is more prominent when the target is to learn multiple downstream tasks. In this work, the authors propose an algorithm, named AMF, which aims to actively fine-tune large-scale pretrained models with the minimum number of demonstrations possible, collected from more than one task. The approach aims to maximize the information gain of demonstrations about the expert policy by selecting the right task for the upcoming demonstration collection. Under some strong assumptions, this work offers theoretical guarantees for matching the performance of the expert policy used for collecting the demonstrations. In addition, the authors offer a practical version of the algorithm based on a GP policy, called AMF-GP, by tackling the occupancy and entropy estimation. Furthermore, a more realistic approach, named AMF-NN, has been proposed when the policy is modeled by a neural network pre-trained on a large-scale dataset. Both algorithms have been empirically evaluated against the naive approach of sampling tasks uniformly for demonstration collection.

**Claims And Evidence:**

- I believe the problem demonstrated in this work is valid. It is important to minimize the number of demonstrations collected for fine-tuning generalist policies on a task. The issue is even more challenging when we need to decide from which task we should collect demonstrations and how many demonstrations we should collect.
- In my opinion, the motivation, claims, and evidence were clear. The flow of this work is consistently good.
- The theoretical discussion strongly supports the claims presented in this work.

**Essential References Not Discussed:**

- I have no references to add.

**Experimental Designs Or Analyses:**

- The benchmarks used in this work are relevant.
- As mentioned, I have a concern regarding the similarity between the tasks in the metaworld benchmark. I think this is an important weakness to highlight. Nevertheless, I appreciate the robomimic benchmark used.
- I have concerns regarding the scarcity of the baselines.
- I am not familiar with the literature of active (fine-tuning) learning, but I believe methods from this domain can be considered as baselines.
- In addition, as stated in this work, the meta-learning literature has already approaches for a similar objective. I think it is important to illustrate the final performance obtained using the proposed approach in comparison to approaches designed for learning to adapt.

**Methods And Evaluation Criteria:**

- The proposed method sounds. I have no comment on the incorrectness of the approach.
- The benchmarks used are convincing. However, I believe that the chosen metaworld tasks are few and too similar. I would have expected to evaluate the proposed approach on the standard Metaworld scenarios (MT10 and MT50).

**Other Comments Or Suggestions:**

- No comments

**Other Strengths And Weaknesses:**

I have to comment on the last paragraph in the discussion section. I believe we already have so many open-sourced generalist policies. In my opinion, the major weakness in this work is the lack of experiments that show the ability of the proposed approach in fine-tuning different large models (other than Octo) for robot learning. I think that benchmarking with more than one backbone model will strengthen the soundness of the proposed approach and validate the claims.

**Questions For Authors:**

- The chosen metaworld scenario consists of only 4 tasks; the tasks, in my opinion, are very similar. Why not consider many tasks from the standard set of MT10 and MT50 in metaworld?

**Relation To Broader Scientific Literature:**

- This work is highly related to the field of robot learning. Recently, many generalist policies have been introduced, and the need for fine-tuning them is high.
- In addition, this work is related to meta-learning and active learning as discussed already by the authors.

**Theoretical Claims:**

- I believe the theoretical claims provided in this work sound. I have checked the claims, and I have no issue with them.

---

> ### Author Rebuttal · Authors · 2025-03-31
>
> We would like to thank the author for their thorough review and positive evaluation of our work. We are happy to provide an answer to each question and comment.
>
> **MT10/50**
> > I would have expected to evaluate the proposed approach on the standard Metaworld scenarios (MT10 and MT50).
>
> The evaluation on Metaworld is designed such that all tasks may share the same initial state distribution. In the standard MT10/MT50 suite, most tasks can be identified through their initial state (e.g., if a button is present, the task is most likely ‘press-button’). This would allow one to potentially quantify policy uncertainty from the states alone, independently of the specified tasks, arguably simplifying the problem.
> In our instantiation, all tasks share the same initial state distribution (faucet and cup are both always present). On one hand, this has the advantage of stressing uncertainty quantification, ensuring that task specification cannot be ignored. On the other hand, a shared state space may facilitate transfer learning, which is an important motivation for multi-task learning as a whole. These are the reasons motivating the adoption of the current Metaworld setup.
>
> AMF however remains fully compatible with disjoint initial state distributions: to demonstrate this, we evaluated AMF on standard MT10 tasks (see new Figure W1 on the [anonymous website](https://sites.google.com/view/active-multitask-finetuning)). Similarly to the existing setup, pre-training only focuses on half of the tasks in mismatched settings, and on all tasks in non-mismatched settings. Results are consistent with those in the current setup, confirming that AMF may be widely applied.
>
> **Other baselines**
> > [...] active (fine-tuning) learning, but I believe methods from this domain can be considered as baselines.
>
> To the best of our knowledge, principled active selection of demonstrations for fine-tuning remains largely unexplored, and we could not find active fine-tuning methods that may be directly applied to this setting. There exists parallel work adopting related objectives, the closest being perhaps [1] in the context of model-based RL. The objective from [1] is however not applicable to this setting: it would involve estimation of a dynamics model from pre-training data, which is not available. Learning the dynamics model on fine-tuning data alone would result in similar behavior to the uniform baseline.
>
> **Other open source models**
> > I believe we already have so many open-sourced generalist policies.
>
> We overall agree with the reviewer: open-source generalist policies are indeed not scarce anymore.
> We have chosen Octo due to its good performance in the Simpler suite (see [2], Figure 7), which should make the evaluation more informative with respect to RT-1-X [3] models. OpenVLA [4] is also available as a base model; however, it was evaluated sparingly in simulation, in which it performed similarly to Octo ([4], Table 12). This motivated our adoption of Octo.
>
> However, we believe that benchmarks remain sparse. The few existing real-to-sim simulated benchmarks are currently designed to mainly support zero-shot evaluations [2], rather than data collection and subsequent fine-tuning. Other simulated environments exist [5], but the real-to-sim gap has not been properly quantified. For this reason, we believe that a comprehensive evaluation of AMF with open-source models would need to go beyond simulation and include extensive hardware experiments. As such, it would lie beyond the scope of this work, which aims at establishing a principled framework for fine-tuning multi-task policies across different parameterizations and environments.
> We are happy to rephrase the discussion in Section 6 to make this clear.
>
> ---
>
> We hope that these clarifications are helpful in supporting experimental choices, considering the motivation and scope of this work. We would be happy to engage in further discussion or answer any additional questions. Thank you again for taking the time to review out work.
>
> **References**
>
> [1] Mehta et al., An experimental design perspective on model-based reinforcement learning, ICLR 2022
>
> [2] Li et al., Evaluating Real-World Robot Manipulation Policies in Simulation, CoRL 2024
>
> [3] Open X-Embodiment Collaboration, Open X-Embodiment: Robotic Learning Datasets and RT-X Models, CoRL 2023 TGR Workshop
>
> [4] Kim et al., OpenVLA: An Open-Source Vision-Language-Action Model, CoRL 2024
>
> [5] Liu et al., Benchmarking Knowledge Transfer for Lifelong Robot Learning, NeurIPS 2023 D&B

---

### Official Review · Reviewer_9SjT · 2025-03-13

**Overall Recommendation:** 3

**Summary:**

*Note: I previously reviewed this paper during an earlier submission cycle. While I acknowledge the authors' efforts to enhance the manuscript, several concerns I raised earlier remain inadequately addressed in this revision. Thus, I incorporated some parts of my prior review, and adjusted the content according to the current submission.*


This paper introduces AMF (Active Multi-task Fine-tuning), an algorithm for efficiently fine-tuning pre-trained "generalist" robot policies to perform multiple tasks. Given a limited demonstration budget, AMF actively selects which tasks to request demonstrations for, aiming to maximize overall multi-task performance. It does this by selecting tasks that yield the largest information gain about the expert policy, focusing on areas where the current policy is most uncertain.

The authors provide theoretical performance guarantees for AMF under regularity assumptions, showing that it converges to the expert policy in sufficiently smooth MDPs. They also demonstrate AMF's effectiveness in practice, applying it to some robotic manipulation tasks with neural network policies. Experiments in simulated robotic environments like FrankaKitchen and Metaworld show that AMF significantly outperforms uniform task sampling, especially when the pre-training data is skewed towards a subset of tasks. The authors also demonstrated that AMF can be applied to off-the-shelf models like Octo, though the improvement over the naive baseline is marginal.

## update after rebuttal
I appreciate the rebuttal from the authors. I still recommend weak acceptance since generally this paper is solid.

**Claims And Evidence:**

I think most of the claims in this submission are supported by evidence.

**Essential References Not Discussed:**

N/A

**Experimental Designs Or Analyses:**

- FrankaKitchen and MetaWorld are relatively simple robotic benchmarks due to their narrow initial state distributions and short task horizons. Future evaluations would benefit from testing on more challenging robotic benchmarks such as RLBench [1], RoboSuite [2], ManiSkill [3], and BiGym [4], which offer greater complexity and variability. While naive BC + MLP/CNN approaches may not be sufficient for solving these benchmarks, many modern imitation learning methods (e.g., ACT, Diffusion Policy) are capable of achieving strong performance given high-quality demonstrations. Therefore, it should be possible to evaluate AMF on those benchmarks if the BC component is replaced with ACT or Diffusion Policy. *It is crucial to demonstrate that the proposed AMF method can also succeed on these more challenging benchmarks, as improving performance on overly simplified tasks provides limited insight*.

- According to Figure 4, the performance of the proposed AMF-NN method is only marginally better than the uniform baseline. This raises concerns about the practicality of the method and the significance of the results, particularly given the relatively simple nature of the tasks.

[1] James, Stephen, et al. "Rlbench: The robot learning benchmark & learning environment." IEEE Robotics and Automation Letters 5.2 (2020): 3019-3026.

[2] Zhu, Yuke, et al. "robosuite: A modular simulation framework and benchmark for robot learning." arXiv preprint arXiv:2009.12293 (2020).

[3] Mu, Tongzhou, et al. "Maniskill: Generalizable manipulation skill benchmark with large-scale demonstrations." arXiv preprint arXiv:2107.14483 (2021).

[4] Chernyadev, Nikita, et al. "BiGym: A Demo-Driven Mobile Bi-Manual Manipulation Benchmark." arXiv preprint arXiv:2407.07788 (2024).

**Methods And Evaluation Criteria:**

Yes, they make sense to me.

**Other Comments Or Suggestions:**

N/A

**Other Strengths And Weaknesses:**

### Strengths

- The paper studies the timely problem of efficiently fine-tuning generalist robot policies, which are becoming increasingly important in robotics.

- The authors provide performance guarantees for AMF under certain regularity assumptions, proving its convergence to the expert policy in smooth MDPs. This adds to the credibility and understanding of the algorithm's behavior.

- AMF demonstrates improvements over uniform sampling, particularly when the pre-training data is biased towards a subset of tasks. This is an advantage as real-world pre-training datasets are sometimes unevenly distributed.



### Weaknesses

- The effectiveness of AMF, especially with neural networks (AMF-NN), hinges on accurate uncertainty estimation. While the proposed loss-gradient embedding approach works well empirically, uncertainty quantification in neural networks remains a challenging open problem. The performance can degrade if the uncertainty estimates are unreliable.

**Questions For Authors:**

See above sections.

**Relation To Broader Scientific Literature:**

AMF combines these two areas. Traditional active learning focuses on selecting the most informative data points to label in a single-task setting. Multi-task learning aims to improve performance on multiple tasks by sharing information between them. AMF extends active learning principles to the multi-task setting, specifically for fine-tuning a pre-trained policy. This is related to works like those by [1, 2] which apply task-directed data selection and active fine tuning, but AMF extends this to sequential decision-making in robotics. Other works have explored multi-task active learning, but not in the context of fine-tuning pre-trained Transformer-based models for NLP [3].

[1] Smith, Freddie Bickford, et al. "Prediction-oriented Bayesian active learning." International Conference on Artificial Intelligence and Statistics. PMLR, 2023.

[2] Hübotter, Jonas, et al. "Transductive active learning: Theory and applications." Advances in Neural Information Processing Systems 37 (2024): 124686-124755.

[3] Rotman, Guy, and Roi Reichart. "Multi-task active learning for pre-trained transformer-based models." Transactions of the Association for Computational Linguistics 10 (2022): 1209-1228.

**Theoretical Claims:**

The theoretical performance guarantees appear reasonable, though I did not verify the full derivation step by step.

---

> ### Author Rebuttal · Authors · 2025-03-31
>
> We thank the reviewer for a (second) evaluation of our submission. We will address each comment individually.
>
> **Additional environments/policy architectures**
>
> > Future evaluations would benefit from testing on more challenging robotic benchmarks
>
> We understand the significance of more challenging benchmarks. In fact, we have closely followed this helpful suggestion from your earlier review: we have included an evaluation in RoboSuite/RoboMimic [1], which ships with human demonstrations for long-horizon tasks up to 700 steps (Figure 6). As suggested, in this case we fine-tune a Diffusion Policy [2], as we found it to outperform ACT in single-task BC. The results are consistent with those on other benchmarks: AMF significantly outperforms a uniform strategy under distribution mismatch, and overall improves data efficiency. Therefore, AMF remains effective in more complex environments, and with fundamentally different policy parameterizations. We thank the reviewer for providing actionable feedback, and suggesting this extended evaluation, which we believe has substantially increased the breadth of our empirical support.
>
> **Uncertainty estimation**
>
> > The effectiveness of AMF, especially with neural networks (AMF-NN), hinges on accurate uncertainty estimation
>
> We agree: uncertainty estimation is at the core of our method, and we acknowledge this limitation in Section 6. Fortunately, we find that loss gradient embedding perform reliably, as confirmed by the new evaluations with a Diffusion Policy in Figure 6. Moreover, AMF is rather agnostic to the particular choice of uncertainty quantification scheme, and we expect future developments to also benefit our framework.
>
> Finally, we would like to thank the reviewer for pointing out additional references, which we would be happy to integrate into our current related works. We of course remain available for further discussion, in particular if any major comment remains standing.
>
> **References**
>
> [1] Mandlekar et al., What Matters in Learning from Offline Human Demonstrations for Robot Manipulation, CoRL 2021
>
> [2] Chi et al., Diffusion Policy: Visuomotor Policy Learning via Action Diffusion, RSS 2023

---

> > ### Comment · Reviewer_9SjT · 2025-04-05
> >
> > Thank you to the authors for their response. I still recommend weak acceptance.

---

> > > ### Author Response · Authors · 2025-04-08
> > >
> > > We would like to thank the reviewer for their feedback.
> > > If possible, we would like to ask the reviewer to further comment on the evaluation in Robomimic with a Diffusion Policy (Figure 6) - does it meet the reviewer's expectations, or could it be further improved?

---

### Decision · Program_Chairs · 2025-05-01

**Decision:**

Accept (poster)

**Comment:**

The paper considers the setting of active task selection for policy finetuning. The paper is easy to read and the results are sound. Overall, I share the same concerns and probably the feelings of the reviewers. The paper is honest in its claims and results, and it is mostly a combination of existing techniques. From my perspective, the paper's novelty is to a certain extent limited, but I still think it is an interesting contribution to the RL community.